

# Skyrmion crystals in the triangular Kondo lattice model

### Zhentao Wang[1,2∘] and Cristian D. Batista[1,3]

**1** Department of Physics and Astronomy, The University of Tennessee,
Knoxville, Tennessee 37996, USA
**2** School of Physics and Astronomy, University of Minnesota,
Minneapolis, Minnesota 55455, USA
**3** Quantum Condensed Matter Division and Shull-Wollan Center,
Oak Ridge National Laboratory, Oak Ridge, Tennessee 37831, USA

## Abstract

We present a systematic study of the formation of skyrmion crystals in a triangular Kondo Lattice model for generic electron filling fractions. Our results indicate that the four-sublattice chiral antiferromagnetic ordering that was reported more than one decade ago can be understood as the dense limit of a sequence of skyrmion crystals whose lattice parameter is dictated by the Fermi wave-vector. This observation has important implications for the ongoing search of skyrmion crystals in metallic materials with localized magnetic moments.

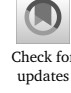

## Contents

---

∘ Present address: Center for Correlated Matter and School of Physics, Zhejiang University, Hangzhou 310058, China

# 1 Introduction

The magnetic skyrmion crystals (SkX) envisioned by Bogdanov and Yablonskii [1,2] were finally discovered in 2009 studying a family of chiral magnets that includes MnSi, $Fe_{1-x}Co_xSi$, FeGe and $Cu_2OSeO_3$ [3–7]. These SkX arise as a superposition of three magnetic spirals produced by competition between ferromagnetic exchange and Dzyaloshinskii-Moriya (DM) interaction [8–10] induced by the non-centrosymmetric structure of these materials. More recently, SkX were also reported in centrosymmetric materials, such as $BaFe_{12-x-0.05}Sc_xMg_{0.05}O_{19}$, $La_{2-2x}Sr_{1+2x}Mn_2O_7$, $Gd_2PdSi_3$, $Gd_3Ru_4Al_{12}$, $GdRu_2Si_2$, $NiI_2$, $Mn_4Ga_2Sn$, and $EuAl_4$ [11–23]. In these cases, the underlying spiral structure arises from competition between different exchange couplings or dipolar interactions [24–28].

To date, most magnetic SkX have been reported in metals, where the magnetic textures act as an effective potential for conduction electrons and reconstruct the electronic bands. Moreover, in the adiabatic limit and in absence of spin-orbit coupling [29], the skyrmion density turns out to be proportional to a fictitious magnetic field that couples to the orbital degrees of freedom of the conduction electrons enabling novel response functions, such as the well-known topological Hall effect [30–33] and the current-induced skyrmion motion [34–37]. Hall conductivities comparable to the quantized value ($e^2/h$) can in principle be achieved if the ordering wave vector of the SkX is comparable to the Fermi wave vector $k_F$. As it was shown in a recent work [38], this condition can be naturally fulfilled in $f$-electron materials where localized magnetic moments interact via Kondo exchange with conduction electrons. In the weak-coupling limit of the Kondo Lattice model (KLM) that is traditionally used to describe these materials, the localized moments interact via an effective Ruderman-Kittel-Kasuya-Yosida (RKKY) interaction mediated by the conduction electrons [39–41]. Since the wave length of the SkX induced by this interaction (in combination with a small easy-axis anisotropy) is $\pi/k_F$, the resulting SkX can produce a very large Hall conductivity (of order $e^2/h$) [38].

One year before the remarkable experimental discovery of SkX, a four-sublattice chiral antiferromagnetic (AFM) ordering was reported to be present in the zero-temperature phase diagram of the triangular KLM (TKLM) [42]. This chiral ordering with uniform spin chirality can be regarded as the dense (short wavelength) limit of SkX with two skyrmions per magnetic unit cell [43]. As it was pointed out in previous works [42,44], the triple-$\boldsymbol{Q}$ four-sublattice chiral AFM ordering is favored relative to a single-$\boldsymbol{Q}$ ordering because it simultaneously gaps out the independent pieces of Fermi surface connected by the symmetry related ordering wave vectors $\boldsymbol{Q}_1$, $\boldsymbol{Q}_2$ and $\boldsymbol{Q}_3$. This argument can in principle be extended to other electron filling fractions that lead to smaller ordering wavevectors $Q = |\boldsymbol{Q}_\nu|$ due to the smaller size of the Fermi surface [see Fig. 1(a)]. Indeed, zero-field SkX with two skyrmions per magnetic unit cell have been reported in TKLMs for smaller filling fractions [45]. More recently, magnetic field-induced SkX with one skyrmion per magnetic unit cell were reported in the low-density and weak-coupling regimes of the KLM on hexagonal lattices by adding a small easy-axis spin anisotropy [38].

Based on these observations, we conjecture that both zero field and field-induced SkX with longer lattice constants (or wavelength) should be rather ubiquitous ground states of the *isotropic* TKLM in the *intermediate-coupling* regime for a continuous range of electron filling fractions [28]. A key observation is that the effective 4-spin interactions (or more generally $2n$-spin interactions with $n > 1$) that are generated in this regime are expected to produce an "attraction" between Fourier components with different ordering wave vectors $\boldsymbol{Q}_\mu$ and $\boldsymbol{Q}_\nu$ with $\mu \neq \nu$. The attractive nature of the effective interaction between different modes reflects the energy gain associated with simultaneously gapping out independent pieces of the Fermi surface.

While our conjecture is supported by an increasing number of experimental results on centrosymmetric metallic magnets [11–23], numerical studies of the KLM have only found SkX for fine-tuned sets of Hamiltonian parameters [45]. According to the results presented in this work, this requirement arises from finite size effects that limit the accuracy of numerical results for the general case. Because of this difficulty, recent theoretical studies of SkX formation have focused on phenomenological models that assume a certain form and sign of 4-spin interactions [46], which are expected to mimic the effective spin-spin interactions generated by the KLM. Since a perturbative treatment of the Kondo exchange breaks down beyond second order at low enough temperatures [44,47], higher order spin interactions are normally incorporated in an ad-hoc manner [46] or by fitting first principles calculations [48,49]. Alternatively, the KLM has also been studied in the double exchange limit [50,51], where additional spin-orbit couplings were introduced to facilitate the stabilization of SkX [52,53].

Only very recently, it was demonstrated that magnetic field induced SkX can naturally emerge in the weak-coupling limit of the KLM for generic filling fractions and model parameters if an easy-axis spin anisotropy [38] or thermal fluctuations [54] are present. In this limit, the KLM can be reduced to the RKKY model [39–41]. Away from this regime, effective $2n$-spin interactions with $n > 1$ become relevant, and the above-mentioned arguments suggest that if the ordering wave vectors connect independent pieces of the Fermi surface, these multi-spin interactions should favor the formation of multi-$\boldsymbol{Q}$ magnetic orderings.

The numerical challenge of obtaining a $T = 0$ phase diagram of the KLM arises from the smallness of the *effective* interactions between localized moments in comparison with the *bare* Hamiltonian parameters. Even the small size effects associated with relatively large finite lattices can alter the relative stability of two competing orders in comparison to the thermodynamic limit. This situation persists for a Kondo exchange interaction $J$ comparable to the nearest-neighbor hopping $t$, posing a serious challenge for numerical techniques that are implemented on finite lattices. Here we avoid these undesirable size effects by implementing a novel variational method *in the thermodynamic limit*. This method reveals that SkX are indeed ubiquitous ground states of the *isotropic* TKLM induced by relatively large multi-spin interactions caused by the above-mentioned Fermi surface effects [28]. The field induced SkX phases emerge above a critical coupling strength $J/t$, indicating that moving away from the RKKY regime of $f$-electron magnets should favor the stabilization these topological spin textures relative to other magnetic orderings.

## 2 Model and results

We consider a 2D TKLM with *classical* local magnetic moments $\boldsymbol{S}_i$:

$$\mathcal{H} = -t \sum_{\langle ij \rangle, \sigma} \left( c_{i\sigma}^\dagger c_{j\sigma} + h.c. \right) + J \sum_{i,\alpha\beta} c_{i\alpha}^\dagger \boldsymbol{\sigma}_{\alpha\beta} c_{i\beta} \cdot \boldsymbol{S}_i - H \sum_i S_i^z + D \sum_i \left( S_i^z \right)^2 , \qquad (1)$$

where the operator $c_{i\sigma}^\dagger$ ($c_{i\sigma}$) creates (annihilates) an itinerant electron on site $i$ with spin $\sigma$, and $t > 0$ is the hopping amplitude of the nearest neighbor bonds. The Kondo exchange $J$ couples the local magnetic moments $\boldsymbol{S}_i$ to the conduction electrons ($\boldsymbol{\sigma}$ is the vector of the Pauli matrices). The last two terms represent a Zeeman coupling to an external field $H$ and an easy-axis single-ion anisotropy ($D \leq 0$). For the centrosymmetric rare-earth based SkX, the moments of the localized spin ($Gd^{3+}$, $Eu^{2+}$) are quite large ($J = 7/2$), which greatly suppresses both the Kondo screening and the quantum fluctuations. With these cases in mind, a classical treatment of the local moments $\boldsymbol{S}_i$ is used throughout this paper, and we use the normalization condition $|\boldsymbol{S}_i| = 1$.

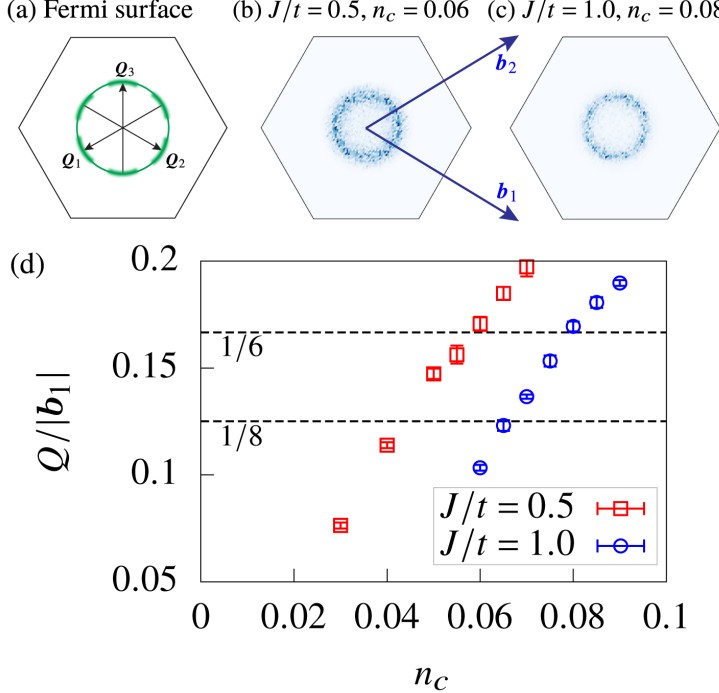

Figure 1: (a) Illustration of independent pieces of Fermi surface that are connected by the symmetry related ordering wave vectors $\{\boldsymbol{Q}_1, \boldsymbol{Q}_2, \boldsymbol{Q}_3\}$ at low filling fractions. (b)-(d) KPM-SLL results for the TKLM on a $96 \times 96$ lattice with $H = D = 0$ and $T = 10^{-5}J^2/t$. (b)-(c) Snapshots of the static spin structure factor $\mathcal{S}(\boldsymbol{q})$. $\{\boldsymbol{b}_1, \boldsymbol{b}_2\}$ are the basis vectors in the reciprocal space. (d) Ordering wave number as a function of the filling fraction.

As we mentioned earlier, the goal of this paper is to study the SkX formation in centrosymmetric metallic magnets for which the KLM provides an appropriate description of the low-energy physics. An important difference between these systems and magnetic Mott insulators deep inside the Mott regime is that effective models of the latter are dominated by short-ranged two-spin interactions, and the stabilization of SkX requires the combination of magnetic frustration and thermal fluctuations or magnetic anisotropy [24–26].

We start by computing the change of the ordering wave vectors $\boldsymbol{Q}_\nu$ as a function of the electron filling fraction $n_c$. In the weak-coupling limit ($J \ll t$), the effective spin-spin interactions are described by an RKKY Hamiltonian and the ordering wave vectors are obtained by maximizing the Lindhard function [38]. However, a different method must be used to find the values of $\boldsymbol{Q}_\nu$ away from this regime because higher order spin interactions produce a significant renormalization of the ordering wave vectors $\boldsymbol{Q}_\nu$.[1]

The values of $\boldsymbol{Q}_\nu$ can be computed in an unbiased manner by iterating two steps at a given temperature. One of the steps "integrates out" the conduction electrons by evaluating the free energy of a proposed spin configuration. The other step consists of a stochastic update the spin configuration based on the value of the free energy. The simplest implementation of these two steps combines exact diagonalization (ED) to compute the free energy and a Metropolis Monte Carlo algorithm to update the spin configurations. However, this combination can only be applied to small lattices because of the $\mathcal{O}(N^3)$ cost of ED [56]. The kernel polynomial method (KPM) [57] replaces the ED step with an approximated polynomial expansion of the single-electron density of states for the given spin configuration. Because the numerical cost

---

[1]See Ref. [55] Fig. (S2) for the dependence of $Q$ on $J/t$ at a fixed filling fraction.

of KPM scales linearly with $N$, the KPM method has been successfully applied to simulations of KLMs [58, 59]. Another numerical improvement based on the KPM method is a stochastic evaluation the local torques (gradients of the free energy) via automatic differentiation, enabling an implementation of Langevin dynamics that greatly speeds up the simulations [60]. The efficiency of the KPM can be further improved by employing a gradient-based probing method [61, 62], that enabled us to simulate KLMs with $\mathcal{O}(10^4)$ lattice sites.

Here we employ a variant of the KPM that utilizes gradient-based probing [60, 62] to obtain an unbiased estimate of the wave vector of low-energy spin configurations on finite lattices of $96 \times 96$ sites. For $H = D = 0$, $T = 10^{-5}J^2/t$ and $J/t = \{0.5, 1.0\}$, we integrate the dimensionless stochastic Landau-Lifshitz (SLL) dynamics with a unit damping parameter using the Heun-projected scheme for a total of 45000 steps of duration $\Delta\tau = 0.5/(J^2/t)$. The order of the Chebyshev polynomial expansion is $M = 1000$ and we use the gradient-based probing method with $S = 256$ colors [62]. The first 30000 steps are discarded for equilibration and the rest 15000 steps are used for measurements. At each temperature, we average over 6 independent runs to estimate the error bars. Finally, the values of $\boldsymbol{Q}_\nu$ can be estimated from the peak positions of the static spin structure factor, defined as

$$\mathcal{S}(\boldsymbol{q}) \equiv \langle \boldsymbol{S}_{\boldsymbol{q}} \cdot \boldsymbol{S}_{-\boldsymbol{q}} \rangle, \tag{2}$$

where $\boldsymbol{S}_{\boldsymbol{q}} \equiv \frac{1}{\sqrt{N}} \sum_i e^{-i\boldsymbol{q}\cdot\boldsymbol{r}_i} \boldsymbol{S}_i$ is the Fourier transform of the real-space spin configurations, and $N$ is the total number of lattice sites.

Figures 1(b)-(c) show typical snapshots of $\mathcal{S}(\boldsymbol{q})$ near the end of the KPM-SLL simulation. For the low filling fractions that we are considering, $\mathcal{S}(\boldsymbol{q})$ takes its maximum value on a ring [55]. Note that, in equilibrium, spin configurations with broken discrete symmetries are generally allowed at finite low temperatures [24]. As we will see later, the lack of symmetry breaking in our KPM-SLL simulation is due to the small energy scale of the effective interactions between local moments in units of the bare interactions of the TKLM.

The magnetic unit cell for each filling fraction can be inferred from the $Q(n_c)$ curve produced by the KPM-SLL simulation, allowing us to exploit the translational symmetry of the optimal spin configuration and find the ground state in the thermodynamic limit. As indicated by the dashed lines of Fig. 1(d), we can always find commensurate ordering wave vectors $\boldsymbol{Q}_\nu(n_c)$ by choosing the right filling fraction. For example, for $Q = |\boldsymbol{b}_1|/L$, the magnetic unit cell contains $L \times L$ spins spanned by the basis $\{L\boldsymbol{a}_1, L\boldsymbol{a}_2\}$, where $\boldsymbol{a}_1$ and $\boldsymbol{a}_2$ are primitive vectors of the triangular lattice.[2]

The $T = 0$ energy density $e$ of each periodic spin configuration is computed by diagonalizing $\mathcal{H}$ in momentum space and integrating the sum of energies of occupied single-particle states over the reduced Brillouin zone $\mathcal{B}_r$:

$$e = \frac{1}{L^2} \sum_{n=1}^{2L^2} \int_{\mathcal{B}_r} \frac{d\tilde{\boldsymbol{k}}}{\mathcal{A}_{\mathcal{B}_r}} \Theta\big[\mu - \epsilon_n(\tilde{\boldsymbol{k}})\big] \epsilon_n(\tilde{\boldsymbol{k}}) + \frac{1}{L^2} \sum_{\boldsymbol{R}} \Big[ -H S_{\boldsymbol{R}}^z + D\big(S_{\boldsymbol{R}}^z\big)^2 \Big], \tag{3}$$

where $\Theta(x)$ is the Heaviside step function, and $\mathcal{A}_{\mathcal{B}_r}$ is the area of $\mathcal{B}_r$.

For convenience, we mainly focus on magnetic orderings with $L = 6$,[3] which correspond to $n_c \approx 0.0586$ for $J/t = 0.5$ and $n_c \approx 0.0796$ for $J/t = 1$ according to Fig. 1. For any fixed parameter set $\{H, D\}$, we minimize the energy density $e$ with respect to $2L^2$ variational parameters (each $\boldsymbol{S}_{\boldsymbol{R}}$ is parametrized by two independent angles) at fixed filling fraction $n_c$. To locate the global minimum, we perform many independent runs with different random initial

---

[2]For $J/t \to 0$, this becomes identical to the susceptibility analysis in Ref. [38].

[3]The choice of commensurate values of $Q$ is only for the convenience of calculation (nearby incommensurate values of $Q$ should produce similar phase diagrams).

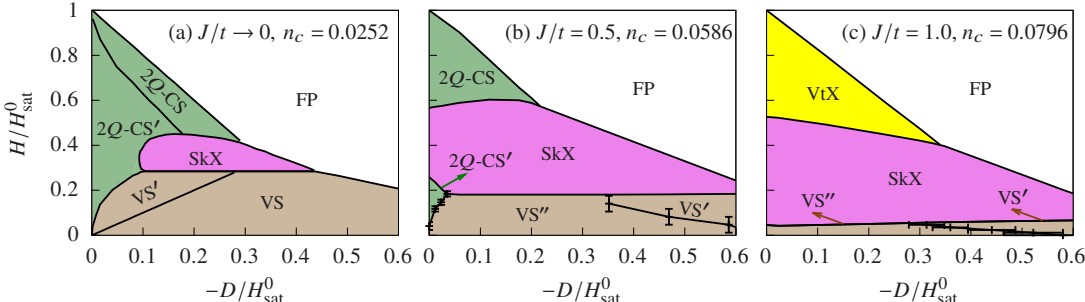

Figure 2: Phase diagrams of the TKLM with easy-axis single-ion anisotropy in a magnetic field with the ordering wave number $Q = |\boldsymbol{b}_1|/6$, where the RKKY limit (a) is taken from Ref. [38]. The error bars of phase boundaries at low field indicate the limited numerical accuracy due to quasi-degenerate states. The saturation field at $D = 0$ is $H_{\text{sat}}^0 = 8.53 \times 10^{-4} t$ for (b) and $H_{\text{sat}}^0 = 4.31 \times 10^{-3} t$ for (c).

spin configurations (typically 20) and keep only the lowest energy solution (see Appendix A for details).

The results of the variational calculation for $J/t = \{0.5, 1\}$ are summarized in Fig. 2. In comparison to the weak-coupling regime ($J/t \ll 1$) described by an effective RKKY Hamiltonian [Fig. 2(b) of Ref. [38], reproduced here as Fig. 2(a)], it is clear that the SkX phase expands upon increasing the Kondo coupling $|J/t|$. This phenomenon highlights the crucial role played by higher order spin interactions [28, 46]. An important consequence of the strengthening of these interactions relative to two-spin interactions is that the single-ion anisotropy is no longer necessary to stabilize the SkX. We have also confirmed that the SkX remains stable for $J/t = 1$ and $D = 0$ when the filling fraction is reduced to obtain a smaller ordering wave vector $Q = |\boldsymbol{b}_1|/8$ (longer wave length).

In the RKKY limit, we found that the optimal wavevector remains practically unchanged for finite $D$ and $H$, indicating that the variational calculation with fixed $L$ remains accurate over the full phase diagram [38]. To verify if this is still true for finite $J/t$, we consider a point inside the $J/t = 0.5$ phase diagram [Fig. 2(b)]: $H = -2D = 0.469 H_{\text{sat}}^0$. The KPM-SLL simulations indicate that a slightly different filling fraction $n_c \approx 0.0548$ (the deviation is comparable to the errorbar of the KPM-SLL simulation) is required to keep the ordering wave vector $Q/|\boldsymbol{b}_1| = 1/6$ unchanged. The variational calculation for the new filling fraction $n_c \approx 0.0548$ at $H = -2D = 0.469 H_{\text{sat}}^0$ confirms that the SkX is still the ground state.

Besides further stabilizing the SkX phase, the larger Kondo exchange also produces a few multi-$\boldsymbol{Q}$ phases that do not appear in the RKKY limit [38]. In particular, a new vertical spiral phase (VS″) [see Fig. 3(a)] and a vortex crystal phase (VtX) [see Fig. 3(c)] appear at low and high fields respectively (see Appendix B for the Fourier analysis of the VS″ and VtX states). We note that the distribution of local scalar spin chirality, $\boldsymbol{S}_i \cdot (\boldsymbol{S}_j \times \boldsymbol{S}_k)$, on each triangular plaquette exhibits chiral stripes with alternating signs (zero net scalar chirality) in the VS″, 2$Q$-CS and 2$Q$-CS′. Similar stripes of scalar chirality have been reported in Refs. [63, 64]. Vortex crystals below the saturation field have also been reported for spin models with short-range anisotropic exchange interactions [65–67].

To date, there are only two direct numerical confirmations of the SkX ground states in the TKLM (1). The first one is the four-sublattice chiral ordering [42, 60, 68, 69] which is strictly speaking the short wavelength limit ($L = 2$ and $Q = |\boldsymbol{b}_1|/2$) of the SkX on a triangular lattice. The second one is directly obtained from the KPM-SLL method by fine-tuning both the third neighbor hopping and the chemical potential [45]. Short wavelength multi-$\boldsymbol{Q}$ chiral spin structures on Kagomé KLM were also reported with fine-tuned Fermi surfaces [70]. The fine-

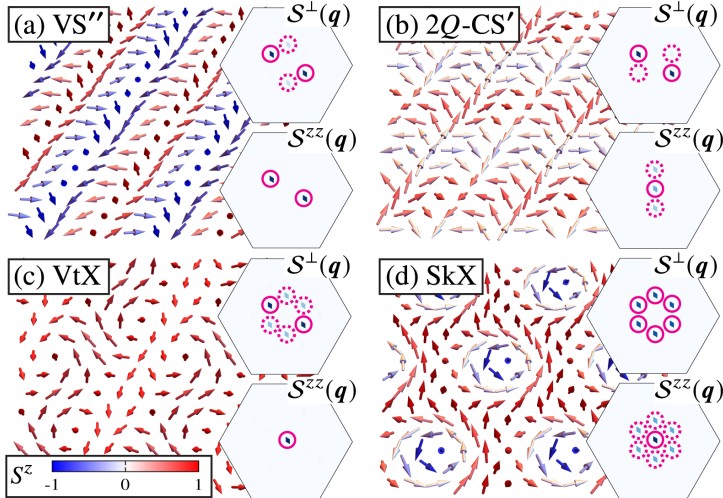

Figure 3: Spin configurations of phases shown in Fig. 2. The insets show the in-plane ($\mathcal{S}^\perp$) and out-of-plane ($\mathcal{S}^{zz}$) static structure factors in the first Brillouin zone. The solid (dotted) circles highlight the dominant (subdominant) peaks. The spin configurations of VS′ and 2$Q$-CS can be found in Ref. [38].

tuning is required to get sharp peaks of the Lindhard function and maximize the magnitude of the effective spin-spin interactions. It is then natural to ask why previous numerical attempts were not able to identify the ubiquitous nature of triple-$Q$ SkX orderings in the TKLM.

The key observation is that the effective spin interactions (RKKY and higher order terms) are much smaller than the bare coupling constant $J$, as it is clear from the values of the saturation fields in Fig. 2 [$H_{\text{sat}}^0 = 8.53 \times 10^{-4} t$ for (b) and $H_{\text{sat}}^0 = 4.31 \times 10^{-3} t$ for (c)]. Furthermore, the splitting between competing single and multi-$Q$ orderings is controlled by an even smaller energy scale that results from the competition between effective higher order spin interactions and the small RKKY energy cost of higher harmonic components of generic multi-$Q$ orderings, which are required to fulfill the normalization constraint $|\mathbf{S}_i| = 1$. This situation leads to very small differences of the energy density, $\Delta e$, in units of the hopping amplitude $t$. Such small energy differences are sensitive to numerical accuracy, which is often limited by system size and by other approximations of the numerical method (e.g., the order of the Chebyshev polynomial expansion and number of random vectors in the KPM-SLL method). To estimate the error introduced by the finite size effects, we simply need to replace the integral in Eq. (3) with a discrete sum on a uniform $(l/L)^2$ grid in $\mathcal{B}_r$. The discrete sum corresponds to the energy density of the given spin configuration on a finite lattice of $N = l^2$ sites. In Fig. 4(a), we consider three converged spin configurations, corresponding to a stable SkX and metastable VS′ and 2$Q$-CS solutions obtained with the variational method for the same parameter set and different random initial spin configurations. The dependence of the energy densities on the linear lattice size, $e_l$, clearly indicates that for $J/t = 0.5$ it is necessary to consider finite lattices with $l \gg 6000$ to obtain results that are representative of the thermodynamic limit. Even For $J/t = 1$, Fig. 4(b) shows that a lattice with $l \gg 1000$ is required to achieve convergence, which is beyond the typical system size that can be reached with the KPM-SLL method. In other words, it is essential to take the thermodynamic limit in Eq. (3) (performing high-accuracy integration) in order to find the correct low energy states of the KLM for general sets of parameters.

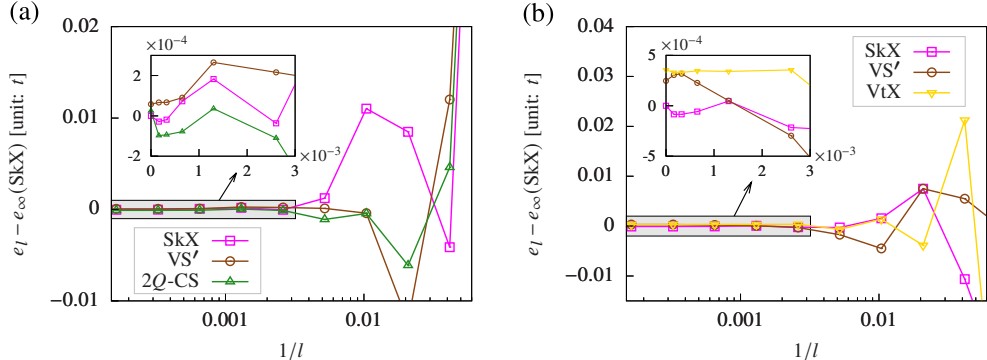

Figure 4: The energy densities $e_l$ of different states evaluated on a uniform $(l/L)^2$ mesh in the reduced Brillouin zone $\mathcal{B}_r$, shifted by the energy density of the SkX state $e_\infty(\text{SkX})$ evaluated by the integral Eq. (3). The data points at $1/l = 0$ are obtained from numerical integration instead of discrete sum. In (a), the three states are obtained by the variational method for the same parameter set $\{J/t = 0.5, n_c = 0.0586, D/t = -10^{-4}, H/t = 3.125 \times 10^{-4}, L = 6\}$, converged from different random initial spin configurations; Similarly in (b), the three states are obtained by the variational method for the same parameter set $\{J/t = 1.0, n_c = 0.0796, D/t = -10^{-3}, H/t = 10^{-3}, L = 6\}$, converged from different random initial spin configurations. For each parameter set, the SkX state is the ground state in the thermodynamic limit, while the other two states are metastable (local energy minimum).

## 3 Conclusion

A key outcome of this work is the confirmation that effective 4-spin and higher order interactions generated in KLMs with hexagonal symmetry lead to SkX that span a large spectrum of wavelengths ranging from $\lambda \gg a$ for small Fermi surfaces [38] to the short wave length or dense limit $\lambda = 2a$ corresponding to the "tetrahedral" ordering reported in Refs. [42, 68, 69]. Based on our results, we conjecture that SkX of intermediate wave lengths between six and two lattice parameters (e.g. $\lambda = 4a$) should also emerge above a critical value of $J/t$ for some range of filling fractions between $n = 0.06$ and $n = 0.25$. We also expect SkX to emerge for even higher filling fractions with more complicated Fermi surfaces. In other words, SkX can be ground states without the need of fine tuning the Fermi surface to produce sharp peaks in the spin susceptibility of the conduction bands. The novel approach that is introduced here avoids this restriction because the results are directly obtained in the thermodynamic limit. Unlike other approaches in which the magnetic unit cell is fixed in an ad-hoc fashion [68], we have used unbiased numerical simulations to determine the optimal magnetic unit cell for a given filling fraction of the conduction bands.

It is worth noting that SkX with both topological charge $n_{sk} = 1$ and $n_{sk} = 2$ per magnetic unit cell were reported before, and the $n_{sk} = 2$ case is often stabilized at zero field [42, 45]. Since the stabilization of both kinds of phases occurs via the generation of 4-spin interactions that favor triple-$\boldsymbol{Q}$ magnetic orderings, our results suggest that SkX of both types should generally emerge for a sequence of electron filling fractions $n_c$ that connect the relatively long ($n_c \ll 1$) and short wave length ($n_c \sim 1$) regimes.

Establishing a *generic* stabilization mechanism of SkX in KLMs is very important because most of the known magnetic SkX have been reported in metals [11–17], where the exchange interaction between magnetic moments and conduction electrons enables novel responses, such as the topological Hall effect (THE) [30–33] and current-induced skyrmion motion [34–

37]. The THE arises from the Berry curvature acquired by the reconstructed electronic bands. A crucial distinctive character of the SkX that emerge in KLMs is that topological Hall effect can be extremely large (comparable to the quantized value) because their lattice spacing is dictated by the Fermi wave vector [15].

Demonstrating the generic nature of the mechanism is particularly relevant because real materials comprise multiple conduction bands and more general forms of Kondo interaction.[4] The key observation is that four and higher-order effective spin interactions generated by processes that involve *independent* pieces of the Fermi surface connected by symmetry related wave vectors favor multi-$Q$ spin configurations [see Fig. 1(a)]. While the amplitude of these effective higher order spin interactions cannot be obtained from perturbation theory because they are non-analytic functions of $J/t$ [28, 44, 45, 47], they can in principle be calculated using other techniques, such as resummation of diagrammatic series. Since their strength relative to the two-spin interaction grows with $J/t$, we expect that, *in absence of easy-axis single-ion anisotropy*, field induced SkX and other multi-$Q$ orderings should emerge above a critical value of $J/t$. This conclusion is supported by an increasing number of experimental results in $f$-electron magnets [71–74]. We note that $d$-electron systems with localized moments in $t_{2g}$ orbitals coupled via Hund's exchange to conduction $e_g$ electrons can also provide natural realizations of the intermediate-coupling regime $J \gtrsim t$ considered in this work [75–80].

## Acknowledgements

We thank Shi-Zeng Lin and Kipton Barros for helpful discussions.

**Funding information** ZW was supported by funding from the Lincoln Chair of Excellence in Physics. During the writing of this paper, ZW was supported by the U.S. Department of Energy through the University of Minnesota Center for Quantum Materials, under Award No. DE-SC-0016371. CDB acknowledges support from U.S. Department of Energy, Office of Science, Office of Basic Energy Sciences, under Award No. DE-SC0022311. This research used resources of the Oak Ridge Leadership Computing Facility at the Oak Ridge National Laboratory, which is supported by the Office of Science of the U.S. Department of Energy under Contract No. DE-AC05-00OR22725.

## A   Variational method

In this section we consider the $T = 0$ commensurate states. Furthermore, we assume that the magnetic unit cell is spanned by the basis $\{L\boldsymbol{a}_1, L\boldsymbol{a}_2\}$, where $\boldsymbol{a}_1$ and $\boldsymbol{a}_2$ are the primitive vectors of the triangular lattice. Note that the value of $L$ can be estimated from unbiased techniques, such as the KPM-SLL method [60,62] used in this paper, which can be implemented in relatively large lattices.

In the following, we will label different sublattices of the magnetic unitcell by $\boldsymbol{R}$, and different unitcells by $\tilde{\boldsymbol{r}}$, so the coordinates of each site can be expressed as the sum $\boldsymbol{r} = \tilde{\boldsymbol{r}} + \boldsymbol{R}$ [81]. The translational symmetry of these commensurate states allows us to perform the Fourier transform:

$$c_{\tilde{\boldsymbol{r}},\boldsymbol{R},\sigma} = \sqrt{\frac{L^2}{N}} \sum_{\tilde{\boldsymbol{k}}} e^{i\tilde{\boldsymbol{k}}\cdot\tilde{\boldsymbol{r}}} c_{\tilde{\boldsymbol{k}},\boldsymbol{R},\sigma}, \tag{A.1}$$

---

[4]While the variational method introduced in this paper can be straightforwardly applied to more realistic cases, the enlarged matrix size due to multiple conduction bands poses an additional numerical challenge, which can be alleviated by using modern parallel computing hardware.

where $N$ is the total number of the lattice sites, and $\tilde{\boldsymbol{k}}$ labels the allowed momenta in the reduced Brillouin zone $\mathcal{B}_r$.

The KLM considered in the main text becomes block-diagonal in $\tilde{\boldsymbol{k}}$-space, $\mathcal{H} = \sum_{\tilde{\boldsymbol{k}}} \mathcal{H}_{\tilde{\boldsymbol{k}}}$, with

$$
\mathcal{H}_{\tilde{\boldsymbol{k}}} = \sum_{\boldsymbol{R}} \left[ -t \sum_{\eta} \sum_{\sigma} c_{\tilde{\boldsymbol{k}},\boldsymbol{R},\sigma}^{\dagger} c_{\tilde{\boldsymbol{k}},\boldsymbol{R}+\boldsymbol{r}_{\eta},\sigma} + J \sum_{\alpha\beta} c_{\tilde{\boldsymbol{k}},\boldsymbol{R},\alpha}^{\dagger} \boldsymbol{\sigma}_{\alpha\beta} c_{\tilde{\boldsymbol{k}},\boldsymbol{R},\beta} \cdot \boldsymbol{S}_{\boldsymbol{R}} - H S_{\boldsymbol{R}}^{z} + D \left( S_{\boldsymbol{R}}^{z} \right)^{2} \right] . \quad \text{(A.2)}
$$

The single-particle eigenstates are obtained by diagonalizing the $2L^2 \times 2L^2$ block matrix of the operator $\mathcal{H}_{\tilde{\boldsymbol{k}}}$ whose eigenvalues are denoted by $\epsilon_{\tilde{\boldsymbol{k}},n}$.

The energy density at $T = 0$ can be written as

$$
e = \frac{1}{N} \sum_{\tilde{\boldsymbol{k}}} \sum_{n=1}^{2L^2} \Theta \left( \mu - \epsilon_{\tilde{\boldsymbol{k}},n} \right) \epsilon_{\tilde{\boldsymbol{k}},n} + \frac{1}{L^2} \sum_{\boldsymbol{R}} \left[ -H S_{\boldsymbol{R}}^{z} + D \left( S_{\boldsymbol{R}}^{z} \right)^{2} \right] . \quad \text{(A.3)}
$$

While Eq. (A.3) is valid for any system size, it is crucial to take the thermodynamic limit $N \to \infty$ to identify the correct ground state of the KLM. This is done by converting the discrete sum $\frac{1}{N} \sum_{\tilde{\boldsymbol{k}}}$ into the integral:

$$
e = \frac{1}{L^2} \int_{\mathcal{B}_r} \frac{d\tilde{\boldsymbol{k}}}{\mathcal{A}_{\mathcal{B}_r}} \sum_{n=1}^{2L^2} \Theta \left( \mu - \epsilon_{\tilde{\boldsymbol{k}},n} \right) \epsilon_{\tilde{\boldsymbol{k}},n} + \frac{1}{L^2} \sum_{\boldsymbol{R}} \left[ -H S_{\boldsymbol{R}}^{z} + D \left( S_{\boldsymbol{R}}^{z} \right)^{2} \right] , \quad \text{(A.4)}
$$

where $\mathcal{A}_{\mathcal{B}_r}$ is the area of the reduced Brillouin zone $\mathcal{B}_r$. For the canonical ensemble used in this paper, the chemical potential $\mu$ is determined self-consistently at every step of the minimization from the filling fraction:

$$
n_c = \frac{1}{2L^2} \int_{\mathcal{B}_r} \frac{d\tilde{\boldsymbol{k}}}{\mathcal{A}_{\mathcal{B}_r}} \sum_{n=1}^{2L^2} \Theta \left( \mu - \epsilon_{\tilde{\boldsymbol{k}},n} \right) . \quad \text{(A.5)}
$$

To this point, the meaning of "variational" becomes clear. The variational space is defined by $2L^2$ parameters corresponding to the two angles of the classical moment $\boldsymbol{S}_{\boldsymbol{R}}$. Equipped with a reliable optimization routine [82], the only control parameter of accuracy is just $L$. In other words, for any integer $L$, if we can pre-compute the corresponding filling fraction $n_c$ exactly, then the energy expression (A.4) also becomes exact. Since we pre-computed the relation of $L$ vs $n_c$ from KPM, the deviation from the exact solution is controlled by the accuracy of KPM when we computed the values of $\boldsymbol{Q}_{\nu}$.

The $T = 0$ states can be obtained by minimizing $e$ at fixed $n_c$ as a function of these $2L^2$ variational parameters [82–84]. For local minimization algorithms, the converged results are often metastable local minima (different initial conditions can lead to different final states). For each parameter set, we typically perform 20 independent runs with different random initial spin configurations. More runs are required to select out the global minimum near the phase boundaries where the competition between different states is more subtle.

The combination of ED, integration and minimization is computationally expensive. To obtain converged results in reasonable amount of time, it is beneficial to use derivative based minimization algorithms. The derivative of the energy density is given by:

$$
\frac{de}{d\boldsymbol{S}_{\boldsymbol{R}}} = \frac{1}{L^2} \int_{\mathcal{B}_r} \frac{d\tilde{\boldsymbol{k}}}{\mathcal{A}_{\mathcal{B}_r}} \mathrm{Tr} \left[ f(\tilde{\boldsymbol{k}}) \frac{dh(\tilde{\boldsymbol{k}})}{d\boldsymbol{S}_{\boldsymbol{R}}} \right] + \frac{1}{L^2} \left[ -H \begin{pmatrix} 0 \\ 0 \\ 1 \end{pmatrix} + 2D \begin{pmatrix} 0 \\ 0 \\ S_{\boldsymbol{R}}^{z} \end{pmatrix} \right] , \quad \text{(A.6)}
$$

where $h(\tilde{\boldsymbol{k}})$ is the $2L^2 \times 2L^2$ matrix of the operator $\mathcal{H}_{\tilde{\boldsymbol{k}}}$ given in Eq. (A.2),

$$f(\tilde{\boldsymbol{k}}) = \sum_{n=1}^{2L^2} \Theta\left(\mu - \epsilon_{\tilde{\boldsymbol{k}},n}\right) |\psi_{\tilde{\boldsymbol{k}},n}\rangle\langle\psi_{\tilde{\boldsymbol{k}},n}|, \tag{A.7}$$

is the density matrix, and $|\psi_{\tilde{\boldsymbol{k}},n}\rangle$ is the eigenvector with eigenvalue $\epsilon_{\tilde{\boldsymbol{k}},n}$.

## B  Fourier analysis

For most of the states that appear in this paper, the Fourier analysis has been documented in Ref. [38]. Here we further analyze the new states that appear in the phase diagram for larger values of $J/t$.

Denote the ordering wave vectors as

$$\boldsymbol{Q}_1 = -\boldsymbol{b}_2/L, \qquad \boldsymbol{Q}_2 = \boldsymbol{b}_1/L, \qquad \boldsymbol{Q}_3 = -\boldsymbol{Q}_1 - \boldsymbol{Q}_2. \tag{B.1}$$

In the VS$''$ phase, the normalized spin configurations $\boldsymbol{S}_r = \boldsymbol{m}_r/|\boldsymbol{m}_r|$ can be parametrized as:

$$m_{\boldsymbol{r}-\boldsymbol{r}_0}^x = -a_1 \cos\phi \sin(\boldsymbol{Q}_1 \cdot \boldsymbol{r}) + a_2 \sin\phi \sin(\boldsymbol{Q}_2 \cdot \boldsymbol{r}), \tag{B.2a}$$

$$m_{\boldsymbol{r}-\boldsymbol{r}_0}^y = -a_1 \sin\phi \sin(\boldsymbol{Q}_1 \cdot \boldsymbol{r}) - a_2 \cos\phi \sin(\boldsymbol{Q}_2 \cdot \boldsymbol{r}), \tag{B.2b}$$

$$m_{\boldsymbol{r}-\boldsymbol{r}_0}^z = a_0 - a_1 \cos(\boldsymbol{Q}_1 \cdot \boldsymbol{r}). \tag{B.2c}$$

The normalized spin configurations of the VtX phase can be parametrized as:

$$m_{\boldsymbol{r}-\boldsymbol{r}_0}^x = a_1 \sin\phi \sin(\boldsymbol{Q}_1 \cdot \boldsymbol{r}) - a_2 \cos\phi \left[\cos(\boldsymbol{Q}_2 \cdot \boldsymbol{r} + \theta) - \cos(\boldsymbol{Q}_3 \cdot \boldsymbol{r} + \theta)\right], \tag{B.3a}$$

$$m_{\boldsymbol{r}-\boldsymbol{r}_0}^y = a_1 \cos\phi \sin(\boldsymbol{Q}_1 \cdot \boldsymbol{r}) + a_2 \sin\phi \left[\cos(\boldsymbol{Q}_2 \cdot \boldsymbol{r} + \theta) - \cos(\boldsymbol{Q}_3 \cdot \boldsymbol{r} + \theta)\right], \tag{B.3b}$$

$$m_{\boldsymbol{r}-\boldsymbol{r}_0}^z = a_0 - a_3 \sin(\boldsymbol{Q}_1 \cdot \boldsymbol{r}). \tag{B.3c}$$

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
