# Peer review of "Skyrmion Crystals in the Triangular Kondo Lattice Model"

_SciPost Physics, doi:SciPost Phys. 15, 161 (2023)_

## Round 2 · Referee Report · Anonymous (Referee 1) · 2022-8-12

Report

The authors numerically studied the ground-state ($T=0$) properties of the triangular Kondo lattice model with emphasis on the skyrmion crystal (SkX) formation by means of the combination of the kernel polynomial method and the variational method, and presented the $T=0$ phase diagram in the easy-axis-type anisotropy vs. magnetic-field plane. They observed that the SkX state was stabilized in a certain region of the phase diagram for generic electron filling, and that the SkX regions tend to be expanded for stronger exchange coupling $J/t$, yielding the stable $T=0$ SkX state even without anisotropy when $J$ becomes comparable to $t$.

The subject is of current interest both theoretically and experimentally, the numerical analysis seems mostly reasonable (see, however, the comment 3 below), and the obtained results are informative to researchers in this and related fields. The referee is basically positive about the paper. Meanwhile, the referee feels that some parts of the manuscript are a bit abmbiguous, or at least not clear enough, and should be better clarified further. So, the referee wishes to suggest the authors to properly take care of the points given below.

  1. In eq.(1), the Hamiltonian of the KLM, is the summation $<ij>$ taken over only nearest-neighboring pairs ? Please specify.

  2. In the abstract, there appears a sentence "... the four-sublattice chiral antiferromagnetic ordering ... can be understood as the dense limit of a sequence of skyrmion crystals whose lattice parameter is dictated by the Fermi-wave vector.", and a similar sentence also in Section 1. The referee is afraid that such a statement might be a bit misleading to the reader, since the four-sublattice chiral antiferromagnetic ordering observed in Refs. [8,9] has a skyrmion number $n_{sk}=2$, while the SkX observed under fields usually has a skyrmion number $n_{sk}$=1. (As far as the referee can see from Fig. 3(d), your SkX is of the skyrmion number $n_{sk}=1$, i.e., distinct from the chiral ordering with $n_{sk}=2$ observed in Refs. [8,9].) Obviously, the two SkX states cannot be connected continuously since the difference is of topological character, as studied in detail in Ref. [16]. Since the "continuation" of the SkX states, the one observed in the present paper under finite fields and the other the four-sublattice chiral spin texture observed in Refs. [8,9] seems to be one of main results of the present paper, you might better give a bit more cautious statement with regard to the above point.

  3. In the part of the variational calculation, you searched for the global minimum by repeating independent runs (typically 20 times) with different spin initial conditions. Since the energy landscape could be rather complex in this kind of problem (as already mentioned in the manuscript), and especially the relevant scale of the energy difference between the competing phases is so minute relative to the leading energy scale of the model, how you can be sure that 20 trials are enough to reach the true global minimum ? You are strongly recommended to give some more justifications here.

  4. In the context of the present research, the following two works might be added in the reference list, though these works did not directly deal with the KLM, but rather the Heisenberg spin model.

  5. A.O. Leonov and M. Mostovoy, Nature Commun. 6, 8275 (2015), where the $T=0$ phase diagram of the short-range Heisenberg model was given in the easy-axis-type anisotropy vs. magnetic-filed plane prior to Ref. [24], which was very much similar to Fig. 2(a) of the present paper.
  6. K. Mitsumoto and H. Kawamura, Phys Rev B 105, 094427 (2022), which dealt with the isotropic Heisenberg model with the standard RKKY interaction associated with the spherical Fermi surface, and found a stable SkX state at $T>0$ even without the magnetic anisotropy. This works also demonstrated that, on varying the filling (or $k_F$), the ordering wave number at $T=0$ took continuously varying values covering a wide range of wave numbers, similarly to your claim in Fig.1.
  • validity: -
  • significance: -
  • originality: -
  • clarity: -
  • formatting: -
  • grammar: -

Author:  Zhentao Wang  on 2022-09-26  [id 2850]

(in reply to Report 1 on 2022-08-12)
Category:
answer to question

The referee writes:

The authors numerically studied the ground-state ($T=0$) properties of the triangular Kondo lattice model with emphasis on the skyrmion crystal (SkX) formation by means of the combination of the kernel polynomial method and the variational method, and presented the $T=0$ phase diagram in the easy-axis-type anisotropy vs. magnetic-field plane. They observed that the SkX state was stabilized in a certain region of the phase diagram for generic electron filling, and that the SkX regions tend to be expanded for stronger exchange coupling $J/t$, yielding the stable $T=0$ SkX state even without anisotropy when $J$ becomes comparable to $t$.

The subject is of current interest both theoretically and experimentally, the numerical analysis seems mostly reasonable (see, however, the comment 3 below), and the obtained results are informative to researchers in this and related fields. The referee is basically positive about the paper. Meanwhile, the referee feels that some parts of the manuscript are a bit abmbiguous, or at least not clear enough, and should be better clarified further. So, the referee wishes to suggest the authors to properly take care of the points given below.

Our response:

We appreciate the referee's positive assessment of our manuscript. Please see our detailed response below.

The referee writes:

  1. In eq.(1), the Hamiltonian of the KLM, is the summation $\langle ij\rangle$ taken over only nearest-neighboring pairs? Please specify.

Our response:

Yes, $\langle ij \rangle$ is taken over nearest-neighbor only, which will be clarified in the updated version of the manuscript. We note that this choice is different from Ref. [16] where both $t_3$ and $\mu$ must be fine-tuned to stabilize the SkX phase. An important conclusion of our manuscript is that fine-tuning of hopping parameters is not necessary for SkX formation, and that the main obstacle in previous attempts of finding SkX for more general sets of Hamiltonian parameters and filling fractions is finite-size effects that persist for very large cluster sizes.

The referee writes:

  1. In the abstract, there appears a sentence "... the four-sublattice chiral antiferromagnetic ordering ... can be understood as the dense limit of a sequence of skyrmion crystals whose lattice parameter is dictated by the Fermi-wave vector.", and a similar sentence also in Section 1. The referee is afraid that such a statement might be a bit misleading to the reader, since the four-sublattice chiral antiferromagnetic ordering observed in Refs. [8,9] has a skyrmion number $n_{sk}=2$, while the SkX observed under fields usually has a skyrmion number $n_{sk}=1$. (As far as the referee can see from Fig. 3(d), your SkX is of the skyrmion number $n_{sk}=1$, i.e., distinct from the chiral ordering with $n_{sk}=2$ observed in Refs. [8,9].) Obviously, the two SkX states cannot be connected continuously since the difference is of topological character, as studied in detail in Ref. [16]. Since the "continuation" of the SkX states, the one observed in the present paper under finite fields and the other the four-sublattice chiral spin texture observed in Refs. [8,9] seems to be one of main results of the present paper, you might better give a bit more cautious statement with regard to the above point.

Our response:

We thank the referee for the careful reading. It is true that the four-sublattice ordering in Refs. [8,9] has skyrmion number $n_{sk}=2$. As pointed out by the referee, there are indeed two types of SkXs that have been reported in Kondo Lattice models: the ones with $n_{sk}=1$ (focus of our paper), induced by finite magnetic field; and $n_{sk}=2$ that are stable even in absence of magnetic field (Refs. [8,9] at short wavelength, and Ref. [16] at longer wavelength). The statement that appears in the introduction is simply a conjecture that applies independently to both the zero field $n_{sk}=2$ and to the finite field $n_{sk}=1$ phases. While it is clear the two types of Skxs cannot be connected because of their different skymion number per magnetic unit cell, the conjecture states that both types of skyrmion phases should emerge for a sequence of filling fractions, $n_c$, that connect the relatively long ($n_c \ll 1$) and short wave length ($n_c \sim 1$) regimes of the model. It is important to note that, without fine-tuning of electron band structure, only short wavelength SkX phases with $n_{sk}=2$ have been reported for the KLM (Refs. [8,9]). Our work demonstrates that longer wavelength field-induced $n_{sk}=1$ phases are also found in the low-density regime. Since the stabilization of both kinds of phases occurs via the generation of 4-spin interactions that favor 3-$Q$ magnetic orderings, we conjecture that both kinds of SkX phases will emerge for different filling fractions ranging from $n_c \ll 1$ to $n_c \sim 1$. We note, however, that the zero-field $n_{sk}=2$ phase may only emerge for certain densities (wave vectors) that minimize the intensity of the higher harmonics required to fullfil the constraint of fixed spin length $|{\bf S}_i|=1$ and above a critical value of $J$ that is also density dependent.

In summary, the key conclusion of our paper is that SkX phases should exist for a series of ordering wave vectors, ranging from long wavelength all the way to the ones with very short wavelength (~2 lattice spacings) without the need of fine-tuning the electron bands. In other words, the $n_{sk}=2$ SkX should also be stable for other wave vectors besides the ones reported in Refs. [8,9,16]. The reason why previous efforts seem to have overlooked this point is mainly due to numerical finite-size effects. We stress that this "generic" nature of SkX phases of the KLM makes the result more appealing for real materials, which are described by more complex Kondo Lattice Models involving multiple bands. Since 4-spin interactions that favor 3-$Q$ orderings arise from processes that connect independent pieces of Fermi surface (see Fig. 1a), the multi-band nature of the more realistic KLMs does not affect the stabilization mechanism.

The referee writes:

  1. In the part of the variational calculation, you searched for the global minimum by repeating independent runs (typically 20 times) with different spin initial conditions. Since the energy landscape could be rather complex in this kind of problem (as already mentioned in the manuscript), and especially the relevant scale of the energy difference between the competing phases is so minute relative to the leading energy scale of the model, how you can be sure that 20 trials are enough to reach the true global minimum ? You are strongly recommended to give some more justifications here.

Our response:

The referee is correct on that 20 trials may not be enough to locate the global minimum. That is indeed the reason why we chose the words typically 20 times in the manuscript. In computing the phase diagrams, we always carefully monitor the change of energy of the lowest states as we vary $H$ or $D$. The competition between different states becomes naturally more relevant near the phase boundaries, where 20 becomes indeed insufficient. Under those circumstances, we typically observe discontinuities of the ground state energy as a function of $H$ or $D$. Correspondingly, more samplings were performed to pin down the phase boundaries. We thank the referee for bringing up this important point that will be clarified in the next version of the manuscript.

The referee writes:

  1. In the context of the present research, the following two works might be added in the reference list, though these works did not directly deal with the KLM, but rather the Heisenberg spin model.

  2. A.O. Leonov and M. Mostovoy, Nature Commun. 6, 8275 (2015), where the $T=0$ phase diagram of the short-range Heisenberg model was given in the easy-axis-type anisotropy vs. magnetic-filed plane prior to Ref. [24], which was very much similar to Fig. 2(a) of the present paper.

  3. K. Mitsumoto and H. Kawamura, Phys Rev B 105, 094427 (2022), which dealt with the isotropic Heisenberg model with the standard RKKY interaction associated with the spherical Fermi surface, and found a stable SkX state at $T>0$ even without the magnetic anisotropy. This works also demonstrated that, on varying the filling (or $k_F$), the ordering wave number at $T=0$ took continuously varying values covering a wide range of wave numbers, similarly to your claim in Fig.1.

Our response:

We thank the referee for pointing out these references, which will be included in the next version of the manuscript.

---

## Round 2 · Referee Report · Anonymous (Referee 2) · 2022-12-13

Report

The authors study the ground state phase diagram of a Kondo lattice model on a triangular geometry with a focus on skyrmions. The local moments are modelled as classical spins, and the model is susceptible to KPM-type numerics (kernel polynomial method). This allows for an accurate computation of the phase diagram.

I agree with the first reviewer that this topic is of current interest. The results seem sound. I also agree that the presentation is not clear enough. While the authors made some efforts in this direction, there is still room for improvements, and several points are unlcear to me. In particular:

1) In the introduction, the authors talk about the relevance of KLMs, but then they proceed and study the limit of classical moments. It should be made clear to which extent this limit is representative.

2) I think that the introduction is written very narrowly and is clearly tailored to experts (already the second sentence switches the focus to details of TKLMs). I think that a somewhat broader introduction would be helpful.

3) I think the methodology should be explained more clearly to non-experts. Both Section 2 as well as Appendix A assume background knowledge of this KPM variant. Can the key idea be summarized in the main text? The "variational" aspect is not explained clearly enough - there only seems to be one cryptic sentence in Appendix A? What are the numerical control parameters? I think that this information would be greatly beneficial to non-experts. This becomes even more important in light of the fact that the authors claim this approach is "novel".

  • validity: -
  • significance: -
  • originality: -
  • clarity: -
  • formatting: -
  • grammar: -

Author:  Zhentao Wang  on 2023-07-24  [id 3831]

(in reply to Report 2 on 2022-12-13)
Category:
answer to question

The referee writes:

The authors study the ground state phase diagram of a Kondo lattice model on a triangular geometry with a focus on skyrmions. The local moments are modelled as classical spins, and the model is susceptible to KPM-type numerics (kernel polynomial method). This allows for an accurate computation of the phase diagram.

I agree with the first reviewer that this topic is of current interest. The results seem sound. I also agree that the presentation is not clear enough. While the authors made some efforts in this direction, there is still room for improvements, and several points are unlcear to me. In particular:

Our response:

We appreciate the referee's positive assessment of our manuscript. Please see our detailed response below.

The referee writes:

1) In the introduction, the authors talk about the relevance of KLMs, but then they proceed and study the limit of classical moments. It should be made clear to which extent this limit is representative.

Our response:

The limit of classical moments is representative when the magnetic moments are large ($S \gg 1$), in which case both the Kondo effect and quantum fluctuations are greatly suppressed. This is indeed the case for the series of centrosymmetric Gd- and Eu-based compounds that host the skyrmion crystal physics. We have clarified this point in the new version of the manuscript.

The referee writes:

2) I think that the introduction is written very narrowly and is clearly tailored to experts (already the second sentence switches the focus to details of TKLMs). I think that a somewhat broader introduction would be helpful.

Our response:

We have expanded the introduction of the manuscript to make it accessible to a broader readership.

The referee writes:

3) I think the methodology should be explained more clearly to non-experts. Both Section 2 as well as Appendix A assume background knowledge of this KPM variant. Can the key idea be summarized in the main text? The "variational" aspect is not explained clearly enough - there only seems to be one cryptic sentence in Appendix A? What are the numerical control parameters? I think that this information would be greatly beneficial to non-experts. This becomes even more important in light of the fact that the authors claim this approach is "novel".

Our response:

We have expanded the text to add more details about the KPM and the variational method.

---

## Round 3 · Referee Report · Anonymous (Referee 2) · 2023-8-8

Report

I thank the authors for their reply. I support publication.

---

## Round 3 · Referee Report · Anonymous (Referee 1) · 2023-8-11

Report

The referee now finds that the resubmitted manuscript has reached the standard of SciPost publication, and wish to recommend its acceptance.

---

## Round 3 · Author Response

Dear Editor,
Thank you for having considered our manuscript entitled "Skyrmion Crystals in the Triangular Kondo Lattice Model" for publication in SciPost Physics. Both referees found our results of current interest and sound, and raised concerns that some parts of the manuscript could be explained in further detail. After incorporating all comments and suggestions, we would like to resubmit the manuscript for further consideration.

Below please find the list of changes that have been introduced in the revised manuscript, along with our detailed response to the referees on the submission page. We are convinced that the manuscript will stimulate further theoretical/experimental findings of SkX for the more general conditions pointed out by our paper.

Yours sincerely,
Zhentao Wang and Cristian D. Batista

---

## Round 3 · List of Changes

1. We have clarified several ambiguous points in the manuscript, including the nearest neighbor hopping <ij>, the number of trials in the variational calculation, and the validity of the classical moment treatment.

  2. We have added discussion to clarify the difference of topological charge $n_{sk}=2$ and $n_{sk}=1$ cases, as pointed out by Referee-1.

  3. We have included several references relevant to the topic, including the ones pointed out by Referee-1.

  4. We have expanded the text for the introduction, the KPM method, and the variational method, as requested by Referee-2.

---

## Editorial Decision

published